# Antimicrobial Peptides against Bacterial Pathogens: Innovative Delivery Nanosystems for Pharmaceutical Applications

**DOI:** 10.3390/antibiotics12010184

**Published:** 2023-01-16

**Authors:** Esther Imperlini, Federica Massaro, Francesco Buonocore

**Affiliations:** Department for Innovation in Biological, Agrofood and Forest Systems, University of Tuscia, 01100 Viterbo, Italy

**Keywords:** antimicrobial resistance, antimicrobial peptide, nanoparticle, drug delivery, pathogen infection, cytotoxicity

## Abstract

The introduction of antibiotics has revolutionized the treatment and prevention of microbial infections. However, the global spread of pathogens resistant to available antibiotics is a major concern. Recently, the WHO has updated the priority list of multidrug-resistant (MDR) species for which the discovery of new therapeutics is urgently needed. In this scenario, antimicrobial peptides (AMPs) are a new potential alternative to conventional antibiotics, as they show a low risk of developing antimicrobial resistance, thus preventing MDR bacterial infections. However, there are limitations and challenges related to the clinical impact of AMPs, as well as great scientific efforts to find solutions aimed at improving their biological activity, in vivo stability, and bioavailability by reducing the eventual toxicity. To overcome some of these issues, different types of nanoparticles (NPs) have been developed for AMP delivery over the last decades. In this review, we provide an update on recent nanosystems applied to AMPs, with special attention on their potential pharmaceutical applications for the treatment of bacterial infections. Among lipid nanomaterials, solid lipid NPs and lipid nanocapsules have been employed to enhance AMP solubility and protect peptides from proteolytic degradation. In addition, polymeric NPs, particularly nanogels, are able to help in reducing AMP toxicity and also increasing AMP loading. To boost AMP activity instead, mesoporous silica or gold NPs can be selected due to their easy surface functionalization. They have been also used as nanocarriers for different AMP combinations, thus synergistically potentiating their action against pathogens.

## 1. Introduction

Antimicrobial resistance (AMR), which takes place when pathogens like bacteria, viruses, fungi, and parasites no longer respond to conventional antibiotics, is usually considered a hard challenge. Recently, the WHO declared that AMR is one of the top 10 global public health threats facing humanity [1]. In particular, pathogens identified with the acronym ESKAPE (*Enterococcus faecium*, *Staphylococcus aureus*, *Klebsiella pneumoniae*, *Acinetobacter baumannii*, *Pseudomonas aeruginosa*, and *Enterobacter* species) are among the most common causes of life-threatening infections acquired in healthcare facilities [2]. In this context, antimicrobial peptides (AMPs) are considered one of the possible options to add a new class of biomolecules to our available weapons to fight these pathogens. AMPs are fundamental components of the innate immune system and they are produced by vertebrates, invertebrates, bacteria, and plants [3,4]. Their actions are exerted both directly against pathogenic microorganisms (bacteria, viruses, and fungi) and/or parasites, and as an immunomodulatory activity to increase host immune responses [5]. AMP sequences and structures are quite diverse among different taxa, although some common properties could be evidenced. They are usually produced as pre-pro-proteins that are cleaved by a specific protease to obtain the mature (biologically active) form of the AMP. Their killing mechanism against bacterial pathogens implicates the interaction with their cytoplasmic membranes with the aim of membrane lysis or of membrane perturbation and entry into the cell pointing to an intracellular target [6,7,8]. Although more than 5000 AMPs have been identified, very few are already on the market as antimicrobial agents (polymyxin and cubicin for example) due to their low in vivo stability and high toxicity; they are, in fact, easily degraded by proteolytic enzymes, especially in the digestive tract or in the intestinal mucosa. Sometimes they are not strictly selective against pathogen cell membranes [9]. Moreover, another problem that needs to be solved is the high cost of their production by solid-phase peptide synthesis compared to conventional antibiotics. To overcome some of these problems, chemical modifications and different systems of delivery have been developed to improve their stability and bioavailability, and to reduce their toxicity [10,11]. In this review, we have decided to focus in particular on the most innovative AMP delivery systems reported in recent papers. New and efficient ways of encapsulating AMPs are fundamental to overcome the gap between the high number of peptides identified each year and the few patents released for this new class of antibiotics. We have selected nanosystems for which a pharmacological application against ESKAPE bacterial pathogens has been successfully tested in vitro and/or in vivo as they are the most promising, thus showing a real potential to become a new drug delivery system in a next future.

## 2. Nanoparticles for AMP Delivery Applied to Pathogen Infections

AMPs can be encapsulated within nanoparticles (NPs), thus improving peptide stability and selective cell/tissue targeting, reducing eventual toxicity and exposure to proteolytic enzymes, and promoting, in general, their delivery and therapeutic efficacy. Additionally, NP delivery represents a suitable strategy against the progress of AMR. A targeted drug delivery, together with a controlled release from NPs, should guarantee the presence of AMP therapeutic doses around the pathogen microenvironment and avoid suboptimal ones which can lead to the selection of resistant microbial mutants without killing the target.

Over the last few years, several nanosystems have been applied for AMPs against bacterial infections (Figure 1) and some of the most representative studies involved in this topic are reported in Table 1. Here, we discuss those nanomaterials used to deliver AMPs, such as lipid carriers, polymer carriers, silica-based NPs, and metal nanosystems which presented antibacterial efficacy against different pathogens. The biological activity of these AMP-NPs will be evaluated in the successive sections of this review.

### 2.1. Lipid-Based Nanoparticles

In the era of pharmaceutical nanocarriers, solid lipid nanoparticles (SLNs) are emerging as a promising delivery system for a variety of therapeutic agents, not only for microbial infections but also for diabetes, neurological disorders, skin diseases, and especially cancer. SLNs are colloidal NPs with a diameter between 50 and 1000 nm and a crystal solid lipid core, stabilized by interfacial surfactants and, eventually, by cosurfactants [12]. They represent an alternative to traditional lipid carriers such as liposomes and micellosomes/micelle-lipid nanocapsules, with many advantages and few disadvantages [13,14]. The peculiar nanocarrier properties of SLNs depend on the composition of their solid core, usually formed by mono-, di-, and triglycerides, fatty acids, fatty alcohols, and waxes [12]. The evaluation of all possible reported SLN formulations is out of the scope of this review, however, it is important to underline that they have to be carefully selected based on the type of pharmaceutical application and the drug’s nature [15]. Being solid at physiological temperature, SLNs are versatile and robust nanocarriers enabling a controlled drug release [16]. Moreover, high biocompatibility and biodegradability are guaranteed, as well as the capacity to encapsulate both hydrophilic and lipophilic drugs, enhancing their stability, solubility, bioavailability, and, lastly, delivery by a wide diversity of administration routes, from oral to nasal, percutaneous, parental, and even ocular [15]. Specifically, oral administration of SLNs can determine an increased drug bioavailability because the encapsulated drugs are protected by strong pH changes and digestive proteolytic enzymes, typically of the gastrointestinal (GI) tract environment [17]. Moreover, also local delivery can be promoted due to a relatively strong adhesion of SLNs to the GI mucosa thus prolonging the drug-target residence times [15]. However, some disadvantages related to low drug loading, due to potential recrystallization processes, or to the possible drug expulsion during SLN storage, need to be considered [12].

SLNs were employed to develop delivery systems for lacticin 3147, a broad-spectrum bacteriocin produced by the food-grade strain *Lactococcus lactis* and active against Gram-positive bacteria, like *Listeria monocytogenes* and *Clostridioides difficile*, which are responsible for infections in the GI tract [18]. However, lacticin use for this clinical application is limited by its physicochemical properties, such as poor aqueous solubility and susceptibility to degradation by human proteases acting in the duodenum [19]. To overcome these limits, an efficient delivery system for lacticin was obtained/developed by Ryan and colleagues [20], where SLNs of uniform size have been loaded with two peptides obtained from the lacticin sequence with an encapsulation efficiency (EE) of about 87%. However, a low efficiency was obtained when lacticin peptides were individually encapsulated into SLNs. Interestingly, the double lacticin peptides-system in the SLN dispersion was more performing in terms of stability, drug loading capacity, and bactericidal activity than the single peptide-occupied one [19,20].

Lipid nanocapsules (LNCs) are used to solve, in general, the problem of drugs’ poor water solubility and for AMP delivery, thanks to the affinity of lipid components for the cell membrane [21]. These biomimetic and highly biocompatible nanocarriers are prepared by using nontoxic and safe components. They are, in fact, formed by an oil core, typically of capric/caprylic acid triglycerides, surrounded by a rigid shell of a hydrophilic surfactant such as macrogol 15 hydroxystearate [22]. In addition to the surfactant, a cosurfactant, such as lecithin, is commonly used to enhance LNC stability and reduce their relative size by preventing the formation of NP aggregates [23]. Recent studies explored the use of fatty acids or monoglycerides, particularly monolaurin (ML), as a cosurfactant, with the great advantage that this component is an antibacterial [24]. In fact, this monoester of lauric acid and glycerol has been proven to make LNCs active against Gram-positive bacilli and cocci, including *Bacillus anthracis* and resistant strains of *S. aureus*, respectively [25]. To obtain an efficient nanosystem, it is important to set up and optimize the composition in terms of lipid/surfactant/cosurfactant percentages. Interestingly, the physicochemical characterization of LNCs containing 46.6% triglycerides, 38.4% macrogol, 15% hydroxystearate, and 15% monolaurin, showed a polydispersity index (PDI) below 0.1 and a hydrodynamic size of about 36 nm, thus suggesting that this formulation, with zeta potential close to neutral, was stable and homogeneously distributed [21]. Such parameters, especially the small size and large surface area, may promote interactions between LNCs and AMPs. Although these interactions, mainly electrostatic, are essentially dependent on the experimental conditions, the surface adsorption of an AMP into LNCs, even without ionizable surfactants, is possible to obtain, thus reaching adsorptions ranging from 34% to 62%, without affecting the LNC size [22]. This possibility is, in fact, an alternative strategy to the covalent attachment and encapsulation used for loading drugs into delivery nanocarriers. However, the drug loading for an LNC is lower compared to hydrophilic polymer-based nanocarriers; in fact, a peptide loading of 1.5–2.7% can be considered optimal for LNCs. In this regard, ML-LNCs loaded with peptide derivatives of plectasin are considered promising for AMP delivery against *Staphylococcus aureus* (SA) and methicillin-resistant SA (MRSA), and for the treatment of skin and soft-tissue infections [22]. Plectasin is 40 amino acids defensin-like cationic AMP isolated from the saprophytic fungus *Pseudoplectania nigrella* [26]. Its derivatives, such as AP114 and AP138, obtained by in silico analysis of the plectasin sequence, showed high in vitro and in vivo activity against MRSA pathogens [22,27]. Unlike other AMPs which act by disrupting bacterial membranes, plectasin derivatives directly bind to the cellular precursor lipid II inhibiting membrane biosynthesis [26]. Interestingly, both plectasin derivatives have been adsorbed on ML-LNCs with an EE ranging from 34% to 62% and, as expected, with no great impact on the NP physicochemical properties. Moreover, the synergistic effect due to the combination of plectasin derivatives with ML-LNCs resulted in a reduction of dosage with a consequent potential decrease in drug toxicity and resistance development [22].

To increase the EE to about 98% (at a drug loading of 0.151%), Groo and colleagues [28] successfully encapsulated plectasin-derivative AP138 into LNC using reverse micelles (RM). This nanosystem was obtained by incorporating a micelle-loaded oil into the LNC lipid core at a temperature above the phase-inversion one. This study showed that the RM system is very promising for the encapsulation of hydrophilic/hydrosoluble peptides into the oily LNC core [28]. Interestingly, the physicochemical analysis confirmed that AP138-RM-LNC is characterized by a stable and homogeneous size distribution. Moreover, this nanosystem also displayed a rapid in vitro AP138 release that is potentially correlated with an in vivo improved antibacterial activity of the AP138-loaded LNCs [28].

### 2.2. Polymeric Nanoparticles

Polymeric NPs have shown great potential as nanocarriers over recent years, due to their physicochemical properties resulting in controllable size and shape production [29]. Additionally, they provide a suitable system to target drugs at the release site of infection, thus avoiding potential toxic systemic effects. These colloidal NPs comprise either natural or synthetic polymers which, in turn, can encapsulate an active compound within their core or adsorb it onto their surfaces. Peculiar characteristics of polymeric particles that render them particularly useful as drug delivery systems are the significant stability of biological fluids and their biodegradability, with a high impact on their safety and biocompatibility [30]. Usually, natural polymers, such as chitosan, dextran, collagen, hyaluronic acid, albumin, gelatin, and alginate are nonsynthetic biodegradable polymers. Moreover, synthetic polymers such as poly(D,L-lactide) (PLA), poly(D,L-glycolide) (PLG), and copolymer poly(lactide-co-glycolide) (PLGA) are also biodegradable. Both natural and synthetic polymers have been used to prepare NPs. Biodegradable polymers are used to overcome the systemic toxicity of drug-loaded NPs and to promote drug release [30]. In this context, Falciani et al. [31] reported a new nanosystem made up of dextran as a promising nanocarrier for the SET-M33 peptide. This non-natural cationic AMP is currently under investigation for the treatment of Gram-negative bacteria pulmonary infections. To perform such a nanosystem, the SET-M33 peptide was captured on a dextran-based single-chain polymer NP (DXT-NP) [31]. Taking advantage of being easily functionalized and of the DXT-NP water dispersion, the covalent binding with mercaptopropionic acid rendered its surface negatively charged in order to favor electrostatic interaction with the positively charged SET-M33 peptide. Zeta potential measurements confirmed the SET-M33 peptide loading of DXT-NP and its physicochemical characterization showed a hydrodynamic small size of 18 nm and an acceptable PDI of 0.3, without any evidence of aggregation [31].

Similarly, chitosan-based NPs are promising as delivery systems for AMP. It is well known that chitosan is a naturally abundant polymer that is nontoxic, biocompatible, biodegradable, and with inherent antimicrobial activity, as a result of electrostatic interactions causing disruption of cell membranes [32]. Recently, Hassan et al. [33] used chitosan NPs as an efficient nanocarrier for the mastoparan peptide against MDR *A. baumannii* clinical isolates. This pathogen is responsible for typically nosocomial infections such as wound and urinary tract ones, septicemia, and ventilator-related pneumonia [34]. Mastoparan, an amphiphilic AMP isolated from wasp venom, contains 14 amino acids which are mainly hydrophobic residues [35]. Like other AMPs, mastoparan acts by disrupting cell membranes with conventional proposed mechanisms (barrel stave, toroidal pore, or carpet models), leading to bacteria death [35]. Interestingly, mastoparan was encapsulated in chitosan-based NP with a loading capacity of 22.63% and an EE of 90%, producing a nanosystem of about 156 nm. This surely represents a successful AMP encapsulation in chitosan-based NPs [33]. Previous studies reported chitosan NPs loaded with other peptides, such as temporin B and cryptidin, reaching average diameters ranging from 100 nm to 190 nm, and an EE < 75% [36,37]. As described below, AMP-loaded chitosan NPs displayed a greater bactericidal effect than peptide or chitosan alone, due to a synergistic action of both components against various clinical bacterial isolates [33].

On the other hand, the potential of polymer-based NPs against mycobacterial infections remains poorly investigated. In this regard, Sharma et al. reported the therapeutic potential of an in silico predicted antimicrobial motif (Pep-H) from the human neutrophil peptide-1 (HNP-1), a human AMP against *Mycobacterium tuberculosis*, following its encapsulation in chitosan-based NPs [38]. Additionally, particle size (244 nm), size distribution (PDI = 0.16), and positive zeta potential indicated that chitosan-based NPs loaded with Pep-H (EE of 72%) were stable and uniformly distributed [38].

Among synthetic polyesters, Poly(lactic-co-glycolic acid) (PLGA) has been extensively investigated as a polymeric NP, especially for its safety and biodegradability. Recent data unravel the potential of PLGA-NPs as a lung delivery system of AMPs against *Pseudomonas aeruginosa* airway infections [39]. In this case, the physicochemical properties, such as NP size, and surface chemistry play a key role in the penetration/diffusion of airway mucus that represents a physical barrier for particles. At the same time, NPs need to protect the encapsulated drug from interaction with extracellular components, thus enhancing drug bioavailability and pharmacokinetics. In this context, Casciaro and colleagues [39] successfully engineered PLGA-NPs loaded with derivatives, a shorter form and a diastereoisomer, of the frog-skin AMP esculentin (Esc). These Esc peptides were previously identified for their antibacterial activity against *P. aeruginosa* [40,41]. Interestingly, the authors wisely coated the PLGA-NPs with a hydrophilic shell of poly(vinyl alcohol) (PVA) to: (i) avoid NP aggregation; (ii) reduce the interaction of PVA-PLG-NPs with airway mucus and extracellular components, and (iii) provide a neutral surface promoting their penetration/diffusion through the lung mucus [39]. The physicochemical characterization of PVA-PLG-NPs loaded with Esc peptides showed an optimal size (<300 nm), a low PDI (<0.061), a slight negative, almost neutral zeta potential, and a satisfying encapsulation of AMPs [39].

### 2.3. Nanogels

Nanogel systems are promising AMP carriers, not yet widely used for antimicrobial applications in contrast to other fields of research such as cancer [42,43]. Nanogels represent attractive drug delivery systems due to their unique properties such as high drug loading capacity and stability, improved cell targeting, and controlled release of drugs in addition to good biocompatibility/biodegradability and reduced toxicity [44,45,46]. Nanogels are cross-linked polymer colloids. They are soft systems, often containing water, which may incorporate large amounts of drugs [47]. In particular, they are composed of a network formed by physical association and chemical cross-linking of amphiphilic polymers [48,49]. There are many polymeric materials used in nanogel preparation and they usually contain hydrophilic segments and hydrophobic side chains; they self-associate into micelle-like domains or interact with the hydrophobic portion of other molecules, thus leading to their incorporation into the hydrophilic nanogel matrix [50]. Contrary to other nanomaterials, described below, such as silica or metal NPs, a nanogel encapsulates and shields the drug, thus resulting in a great advantage for AMPs with regard to their antimicrobial efficiency. In this context, Klodzinska and colleagues [44] reported the first formulation of biopolymer nanogels incorporating a lysin-based peptidomimetic (LBP) with potential antibacterial activity against Gram-negative bacteria, mainly *P. aeruginosa*. These peptidomimetics incorporating unnatural amino acids represent those strategies that have aimed to improve the bioavailability and stability of AMPs. Previous studies, in fact, showed the high potential of these LBP series, where their length plays a key role in determining the antibacterial activity [51,52]. The most promising peptidomimetic has been incorporated into a nanogel formulation using an octenyl succinic anhydride-modified hyaluronic acid (HA) polymer [44]. The most promising nanogel formulation, in turn, was obtained by modulating the peptidomimetic to polymer ratio, thus reaching a high EE (88%) of LBP, a small particle size (175 nm), and a negative zeta potential together with reduced cytotoxicity, while maintaining a potent antimicrobial efficacy against *P. aeruginosa* [44].

Recently, Fasiku and colleagues [53] prepared an HA-based nanogel for the codelivery of nitric oxide (NO) and a novel synthetic AMP (RKKKKLLRKKC) with an improved antibacterial activity due to its high content of L and K amino acid residues. Alternatively, considering the intrinsic antibacterial and antibiofilm properties of NO, donor molecules of NO, such as S-Nitroso-N-acetyl-DL-penicillamine (SNAP), are generally used to increase NO’s short half-life and stability, aspects that could limit its antimicrobial applications [54]. Interestingly, this study first combined the delivery of an AMP with SNAP to synergistically potentiate their antibacterial and antibiofilm activities against Gram-positive and Gram-negative bacteria [53]. In this regard, a successful nanogel formulation was achieved by cross-linking, under alkaline conditions, the –OH group of HA with divinyl sulfone, thus obtaining sulfonyl bis-ethyl crosslinks and a completely formed nanogel system [53]. The physicochemical characterization of HA-SNAP-AMP nanogel showed an average size (>580 nm), a PDI (>0.4), and a zeta potential (>−30 mV) similar to those typical of hydrogels. Moreover, these characteristics are in line with the possibility of reaching high drug load capacity and, as a consequence, a nanogel porosity compatible with an extended release of incorporated antimicrobials [53].

Among new nanogel preparations, stimuli-responsive nanogels are emerging because they may be designed to swell or deswell in response to temperature, pH, or ionic strength variation, thus ensuring selective targeting and controlled drug release [55]. Such a thermosensitive or pH-responsive nanogel has been employed for the delivery of anticancer drugs [56,57]. Recently, a responsive nanogel was reported for the delivery of nisin, an AMP from *Lactococcus lactis* with antimicrobial activity against Gram-positive bacteria [58]. The nanogel system was prepared by using chondroitin sulfate (CS) as a pH-sensitive copolymer and poly(L-lactide) (PLLA) as a thermosensitive moiety, thus giving a dual responsive self-assembled nanogel, loaded with nisin. Interestingly, CS is a sulfated linear anionic polysaccharide that is the most abundant glycosaminoglycan of the extracellular matrix, and it is present on the cell surfaces of many soft tissues. It can form self-assembled structures with another biopolymer, like PLLA, to form a nanogel [58]. In particular, PLLA-graf-CS (PLLA-g-CS) copolymers were synthesized by ring-opening polymerization of L-lactide monomers: nisin was loaded at 37 °C and 42 °C [58]. The most promising nanogel formulation was the one obtained at 37 °C, due to its optimal physicochemical properties, especially with regards to the average size (181 nm), the PDI values (0.31), and the zeta potential (−8.6) [58].

In addition to nanogel, there are also reports of microgels responsive to various stimuli used to release drugs in response to specific diseases [59,60]. As for AMP delivery, the great advantage of microgel systems is their ability to load very high peptide amounts. However, the design and setting up of these formulations are not always easy and successful due to the peptide loading capacity and to its release from the microgel that depends on several factors related to peptide physicochemical characteristics (length, amino acid sequence, charge, and hydrophobicity) [61,62,63]. Despite this problem, Nordstrom and colleagues [42] found that anionic poly(ethyl acrylate-co-methacrylic acid) microgels are able to incorporate considerable amounts of the cationic cathelicidin-derived AMP LL-37 and the immunomodulatory peptide DPK-060, as confirmed by the negative zeta potential of the loaded nanosystems. As a further support, the authors did not observe any adsorption of microgel systems (empty and peptide-loaded) on bacterial membrane models. On the other hand, it is noteworthy that it is possible to enhance AMP release by acting on peptide length and microgel charge density, particularly by decreasing both these factors [42].

### 2.4. Silica-Based Nanoparticles

Since silica-based materials were approved by the Food and Drug Administration, mesoporous silica nanoparticles (MSNs) containing the porous honeycomb-like structure of silica (SiO_2_) have been deeply investigated, especially as a drug delivery system for cancer therapeutics [64,65]. The peculiar advantages of MSNs are related to their simple synthesis and easy surface chemistry that promote large-scale production [66]. The unmodified silica surfaces are negatively charged due to the presence of the hydroxyl group of tetraethyl orthosilicate. In addition to high physicochemical stability and biocompatibility, easy surface functionalization can be achieved thanks to their high surface area and large pore volume [66]. MSNs are also widely used for efficient drug loading in their core or onto their surface through either electrostatic/hydrophobic interactions or covalent binding, as well as for the possibility to perform controlled drug release and targeted drug delivery [67,68].

In the antimicrobial field, Zhao and colleagues [69] employed MSNs to deliver an Arg-mutant of human defensin 5 (T7E21R-HD5) that is more resistant to enzymolysis and as a more effective action in saline solution than the natural AMP (HD5), and, thus, it is active against intestinal bacterial infections. Interestingly, the physicochemical characterization of MSN-T7E21R-HD5 confirmed its spherical shape with a size of about 60 nm and a negative surface charge supporting the electrostatic attraction of the cationic HD5 Arg-mutant, whose secondary structure remained unmodified after its adsorption on MSN [69]. To protect the T7E21R-HD5 peptide through the oral route and to delay its release in the highly acidic gastric environment, it is noteworthy the development of a natural coating based on succinylated casein (SCN) that is specifically degraded only by intestinal protease [69]. Hence, the use of this promising and biocompatible MSN-SCN could be enlarged to the oral delivery of other enteric drugs.

On the other hand, MSNs can be used as nanocarriers for different combined antibiotics in order to boost the antimicrobial efficacy of those currently available and to simultaneously fight infections caused by different bacteria, especially those requiring high doses of drugs. In this regard, Gounani et al. [70] developed a dual-delivery nanosystem using bare and carboxyl-modified MSNs loaded with two antibiotics, polymyxin B and vancomycin, thus achieving antimicrobial efficiency against both Gram-positive and Gram-negative bacteria. Interestingly, the authors observed dose-dependent adsorption of both antibiotics on MSNs, with a preference and higher affinity for polymyxin B, if compared to vancomycin. As expected, due to the negative charge surface of both MSNs, the adsorption of positively charged antimicrobials was driven by electrostatic interactions. Moreover, the loading capacity of carboxyl-modified MSNs was lower compared to bare MSNs, likely for their smaller surface area and pore size [70].

In a more recent study, silica particles were used to fabricate micro- and nanomotors whose bioactive motion is driven by catalysis of the enzyme urease anchored with a glutaraldehyde linker on the amino groups of the bare silica [71]. These micro- and nanomotors were used to deliver, directly to the infection site, two cationic AMPs particularly susceptible to protease degradation, namely the well-known and previously described LL-37 peptide and the synthetic K7-Pol, isolated from wasp venom and active against a broad spectrum of pathogens [71]. The choice of MSNs as a base material for these bioactive micro- and nanomotors is surely related to their biocompatibility and easy surface functionalization. Following this step, as expected, the authors observed an AMP dose-dependent increase of the nanomotor zeta potential that is consistent with the net positive charge of both tested peptides and the specific characteristics of the surrounding silica particles. This study opens the possibility of the employment of bioactive nanomotors as a therapeutic strategy against microbial infections [71].

### 2.5. Metal Nanoparticles

Among metal-based nanosystems, gold NPs (AuNPs) are emerging as a promising drug delivery system due to their small size, large chemical stability and inertness, large surface area, and, consequently, high drug loading capacity, low cytotoxicity, and biocompatibility [72]. Despite these optimal premises, the studies reporting AuNPs loaded with AMPs are yet still limited. In addition to chitosan-based NPs, Sharma et al. [38] reported, within the same study previously mentioned, the use of AuNPs for Pep-H delivery against mycobacterial infections. The physicochemical analysis confirmed a spherical shape and a very small average size (about 20 nm) for Pep-H-AuNPs and also the stability of the peptide-NP bond, without any particle agglomeration and with minimal peptide release [38]. To the best of our knowledge, this study remains the first paper reporting AuNP loaded with an AMP against *Mycobacterium tuberculosis*.

The surface of these promising nanocarriers (AuNPs) could be easily modified to improve drug delivery properties. In addition to conventional surface modifications such as polyethylene glycol (PEG) or cationic coating, the conjugation of AuNPs with a DNA aptamer (AuNP-Apt), namely a synthetic nucleic acid molecule with a high binding affinity towards specific targets, was reported [73]. As for AMPs, Lee and colleagues [74] developed an AuNP-Apt to deliver HPA3P, a *Helicobacter pylori*-derived peptide, against *Vibrio vulnicus* infections, the leading cause of death due to seafood contaminations. In particular, an AuNP-Apt^His^ conjugate was loaded with HPA3P after His-tagging by simple mixing and incubation at room temperature. Hence, the advantages of AuNPs in terms of low toxicity/immunogenicity meet those of DNA aptamers related to efficient delivery and drug stability. As reported by Lee et al. [74], binding capacity assays showed that 50–60% of HPA3P^His^ associates with AuNP-Apt^His^. Moreover, the physicochemical analysis demonstrated that the complex formation increases both the average size (of one magnitude order, up to about 880 nm) and the zeta potential (up to −25 mV), thus decreasing the surface negative charge and promoting the cellular uptake of the nanoconjugate.

**Table 1 antibiotics-12-00184-t001:** The most representative studies reporting innovative NPs as delivery systems for AMPs against bacterial pathogens.

NP	AMP	Pathogen/Infection	References
Solid lipid nanoparticles (SLNs)	Lacticin 3147	*Listeria monocytogenes*, *Clostridioides difficile*/gastrointestinal infections	[19,20]
Lipid nanocapsules (LNCs)	AP114 and AP138, derived from plectasin	*Bacillus anthracis*, methicillin-resistant *Staphylococcus aureus*/respiratory infections	[21,22]
Lipid nanocapsules (LNCs)	AP138, derived from plectasin	*Staphylococcus aureus*, methicillin-resistant *Staphylococcus aureus*/skin, and soft-tissue infections	[28]
Dextran-based single-chain polymer NP (DXT-NP)	SET-M33 synthetic peptide	Gram-negative bacteria/pulmonary infections	[31]
Chitosan-based polymeric NPs	Mastoparan	*Acinetobacter baumanii*/nosocomial infections	[33]
Chitosan-based polymeric NPs	Pep-H, derived from human neutrophil peptide-1	*Mycobacterium tuberculosis* infections	[38]
Poly (lactic-co-glycolic acid) (PLGA) polymeric NPs	Esculentin-derived peptides	*Pseudomonas aeruginosa*/lung infections	[41]
Hyaluronic acid (HA)-based nanogels	Lysin-based peptidomimetic (LBP)	*Pseudomonas aeruginosa*/lung infections	[44]
Hyaluronic acid (HA)-based nanogels	Synthetic peptide (RKKKKLLRKKC)	*Staphylococcus aureus*, *Escherichia coli*, *Pseudomonas aeruginosa*/various bacterial infections	[53]
Chondroitin sulfate (CS)-based nanogels	Nisin	*Staphylococcus aureus*, *Escherichia coli* infections	[58]
Poly(ethyl acrylate-co-methacrylic acid) microgels	LL-37 derived from cathelicidin and DPK-060 synthetic peptide	Methicillin-resistant *Staphylococcus aureus*, *Escherichia coli*, and *Pseudomonas aeruginosa* infections	[42]
Mesoporous silica nanoparticles (MSNs)	T7E21R-HD5 derived from defensin 5	Multidrug-resistant (MDR) *Escherichia coli*/intestinal infections	[69]
Mesoporous silica nanoparticles (MSNs)	Polymyxin B and vancomycin	Gram-negative and Gram-positive bacterial infections	[70]
Micro- and nanomotors	LL-37 derived from cathelicidin and K7-Pol synthetic peptide	Gram-negative and Gram-positive bacterial infections	[71]
Gold NPs (AuNPs)	Pep-H, derived from human neutrophil peptide-1	*Mycobacterium tuberculosis* infections	[38]
Gold NP-DNA aptamer (AuNP-Apt)	HPA3P derived from Hp(2-20) peptide	*Vibrio vulnificus*/gastrointestinal infections	[74]

## 3. Antimicrobial Activity of Selected AMP-NPs

### 3.1. Lipid-Based Nanoparticles

The encapsulation of lacticin 3147 into SLNs enhanced its antimicrobial activity against *L. monocytogens* ATCC1916 and resistance to protease, similar to α-chymotrypsin, compared to the free peptide [19]. In addition, the peptide release from SLNs has been investigated, evaluating its effect on the microbial growth of *L. monocytogens* ATCC1916 for 48 h. A strong growth reduction was evidenced, especially at 24 h [20].

Similarly, Umerska et al. [22] reported that the inclusion of two peptide derivatives from plectasin, AP114 and AP138, in ML-LNCs improved their antimicrobial activity against *S. aureus* strains. Specifically, a synergistic action of both AP114 and AP138 with ML was observed against different bacterial targets through signal transduction systems. This action resulted in deformations of the bacterial cell shape and disruption of the cell wall and membrane with consequent leakage of cytoplasmic content. It is noteworthy that both peptides, loaded in ML-LNCs, displayed a bactericidal effect on a methicillin-sensitive *Staphylococcus aureus* strain after 18 h and 24 h, with a growth reduction of about eight logs compared to the free peptides or bare ML-LNC [22]. Moreover, the proteolytic stability of the peptide AP138 loaded in RM-LNCs has been investigated using different enzymes, like elastases and trypsin [28]. In particular, the stability of AP138 to trypsin cleavage was enhanced by the peptide inclusion into NP [28].

These studies show the importance of lipid-based NPs applied to AMP delivery, as they are able to limit proteolytic degradation and protect the peptides from an unfavorable environment. These NPs, in fact, are stable when incubated in the presence of artificial gastric (with pepsin) and intestinal (with pancreatin) fluids to mimic their residence time in the stomach and the small intestine [20].

### 3.2. Polymeric Nanoparticles

Among polymeric nanoparticles suitable for AMP delivery, Falciani et al. [31] used dextran-based NP (DXT-NP) to encapsulate the SET-M33 peptide for the treatment of pulmonary infection. They showed that SET-M33, either captured on DXT-NP or as a free peptide, had similar activity against the *P. aeruginosa PAO1* strain. Specifically, high antimicrobial activity was observed, at the MIC value, for both free and encapsulated peptides, without bacterial regrowth during 24 h of exposition, as observed by time kill kinetic studies [31]. Regarding the cytotoxic activity, it has been tested both against several human epithelial cells and mouse macrophages. In particular, the encapsulated SET-M33 peptide was slightly more toxic compared to the free peptide, with cell viability values ranging from 65% to 79% at the used concentration of 32 μg/mL, which was twice the MIC value [31]. On the other hand, the authors studied the in vivo activity of both the peptide and the delivery nanosystem in a pneumonia model, performed by infecting a BALB/c mouse with *P. aeruginosa PAO1*. This analysis showed that free and DXT-NP-encapsulated SET-M33 totally eradicated lung infection when used at a concentration of 5 mg/kg. It is noteworthy that DXT-NP improved the lung persistence of the SET-M33 peptide from 1.21 h to 13 h in comparison to the free peptide. Moreover, this nanosystem also affected the metabolic fate of the peptide. It was mainly eliminated through the gastrointestinal tract when loaded into DXT-NP, and through renal excretion when it was in the free form [31]. The in vivo data confirms the importance of the right delivery system choice to increase residence time at the target site of infection.

Another type of polymeric NP applied to AMP delivery is the chitosan-based one. This was used by Hassan et al. [33] as a nanocarrier of the mastoparan peptide against *A. baumanii* infections. The authors observed that the MIC values were lowest for the peptide encapsulated in chitosan-based NP in comparison with the free peptide or the chitosan alone. Moreover, scanning electron microscopy analysis showed that bacteria cell treatment with mastoparan into a chitosan-based NP induces the loss of membrane integrity and the formation of extracellular thread-like structures around the cells. Surprisingly, this phenomenon was not observed for the cells treated with chitosan-based NPs only. In this case, bacterial cells maintained an intact membrane [33]. The encapsulation of mastoparan into the chitosan-based NP also reduced the hemolytic and cytotoxic activities of the peptide. In fact, the free mastoparan showed a dose-dependent hemolytic activity that was not observed when it was loaded into chitosan-based NP, nor at the highest tested concentration. Concerning cytotoxic activity, 98% of rhabdomyosarcoma cell viability was observed when they were treated with mastoparan loaded in chitosan-based NPs; whereas only 42–56% of cells were viable when treated with the free peptide at a concentration of 20 μg/mL [33]. This strongly supports the utility of delivery nanosystems to reduce AMP toxicity. Moreover, as a further confirmation of the in vivo designed nanosystem safety, Hassan et al. [33] tested the mastoparan encapsulated into chitosan-based NPs in a mice BALB/C model infected with *A. baumannii*. They observed a significant reduction in colony count in mice treated with the NP-loaded peptide compared to the control groups (treated with free peptide or bare chitosan NP), thus demonstrating the in vivo stability and activity of mastoparan encapsulated into polymeric NPs [33].

Sharma et al. [38] utilized chitosan-based NPs as nanocarriers of a synthetic peptide (Pep-H), derived from human HNP-1, against the pathogen *M. tuberculosis* [38]. In particular, they observed that the encapsulation of the peptide at a concentration of 0.5 μg/mL is able to reduce the mycobacterial growth in monocytes by 80%; whereas, at the same peptide concentration, the growth inhibition was only by 12% when the monocytes were treated with the free peptide [38]. Moreover, the release of Pep-H from chitosan NPs in simulated intestinal fluids (SIFs) and in phosphate buffer (PB) was very fast only within the first hour of treatment, then it proceeded more slowly. Specifically, 50% of peptide release was observed after 72 h in SIFs and after three days in PB [38]. The hemolytic and cytotoxic activities of encapsulated Pep-H against monocytes were very low, thus indicating the high cytocompatibility of the nanosystem designed for this peptide [38]. Taking these results into account, chitosan is a delivery system of high interest for future medical applications.

In the same context, Casciaro et al. [39] developed a nanosystem for AMP lung delivery against *P. aeruginosa* pulmonary infections. The used nanosystem, in this case, was PLGA, a synthetic polymer, and the encapsulated short-size peptides derived from Esc, Esc(1-21) and its diastereomer Esc(1-21)-1c. Specifically, the authors replaced the amino acids L-Leu and L-Ser with the corresponding D amino acids to increase in vivo stability and reduce cellular aspecificity. They studied the in vitro kinetics release in the PB of the peptides from PLGA-NPs, thus showing that they were initially released very quickly, with 60% of the peptide that is released in the first 3 h and, successively, there was a controlled release phase that lasted three days [39]. The authors also demonstrated the absence of any interaction of the nanoformulation both with mucin and alginate, the principal components of mucus from cystic fibrosis patients, and with the extracellular matrix of *P. aeruginosa* [39]. In fact, they observed that the peptides encapsulated into PLGA-NPs could quickly spread through the artificial mucus and the simulated bacterial extracellular matrix. On the contrary, the free peptides remained partially trapped in these simulated matrices [39]. The inclusion of Esc peptides in PLGA-NPs also improved the in vitro duration of the antimicrobial action against *P. aeruginosa*. Specifically, the antimicrobial activity of free peptides was strong in the first 24 h and decreased in the following 48 h. A growth inhibition of about 97% was observed after 24 h of treatment and 38% after 72 h. The antibacterial activity of Esc-PLGA-NP peptides was less in magnitude compared to the free peptide but remained constant over time. A 60% bacterial growth inhibition was revealed after both 24 h and 72 h [39]. Additionally, the authors demonstrated the safety and effectiveness of Esc-PLGA-NPs peptides in a mouse model of acute *P. aeruginosa* lung infection. In fact, a 17-fold and 4-fold reduction in the number of *P. aeurigonsa* infected cells in the lung was observed using Esc(1-21)-1c and Esc(1-21)-loaded PLGA-NP, respectively [39].

### 3.3. Nanogels

Klodzinska et al. [44] developed different types of nanogel systems based on octenyl succinic anhydride-modified HA polymer that reduced the cytotoxic activity of peptidomimetic, such as LBP, against mammalian cells, such as human hepatoma cancer cells (HepG2) [44]. It is noteworthy that the incorporation of this peptide in nanogels did not change its antimicrobial activity against *P. aeruginosa* PA01 [44].

Another type of HA-based nanogel has been developed and investigated by Fasiku et al. [53]. In particular, they showed that this system, made by codelivering both NO and a synthetic peptide, is not cytotoxic for mammalian cells and that the utilized AMP acts in synergy with NO against *E. coli*, *P. aeruginosa*, and MRSA. Specifically, the combined antimicrobials showed 80% and 82% of biofilm eradication when tested against MRSA and *P. aeruginosa*, respectively [53].

As innovative nanosystems, Ghaeini et al. developed a stimuli-responsive nanogel based on PLLA and CS copolymers loaded with nisin [58]. Specifically, cytotoxic activities of this complex, free peptide, and bare nanogel were studied against human dermis fibroblast cells (HDFCs). Interestingly, the authors showed that the inclusion of nisin into nanogel reduces its cytotoxicity. This phenomenon could be due to the presence of CS in the nanogel, which induces cell proliferation. Moreover, its antimicrobial activity was studied against *E. coli* and *S. aureus* by using the agar well diffusion method. The results evidenced that the growth inhibition zones due to the peptide alone and to the peptide included in nanogel were comparable [58]. It is noteworthy that the nanogel does not modify the antimicrobial activity of nisin, however, at the same time it reduces its cytotoxic effect [53]. These important inputs suggest that the specific nanogel used in this work could be used successfully for in vivo delivery of AMPs. Moreover, Ghaeini et al. [58] also studied the release of nisin by nanogel at different pH (5.4 and 7.2) and temperature (37 °C and 42 °C) values. In particular, at an acidic pH and both 37 °C and 42 °C, the nisin release was higher than that at physiological pH. Here, instead, there was a different release pattern at 37 °C and 42 °C (25% versus 63% of nisin released, respectively). These data confirmed the pH and temperature sensitivities of the developed nanogel and its good potential for pharmaceutical applications [58].

In addition to nanogels, anionic polyacrylic microgels have been tested to encapsulate AMPs, in particular LL-37 and DPK-60 [42]. With regard to DPK-60-loaded nanogel, the authors reported that its antimicrobial activity against *E. coli* and MRSA is similar to that observed for the free peptide, whereas the specific MIC value significantly improved against *P. aeruginosa*. Specifically, a higher MIC decrease was observed for a bacterial clinical *P. aeruginosa* strain rather than for the ATCC correspondent one. Moreover, there was not any particular effect on the hemolytic activity. In fact, the hemolysis percentage was low (about 20%) either for the encapsulated or the free peptide, even at the highest peptide-tested concentration (200 mΜ) [42]. Regarding the LL-37 peptide, its incorporation into a microgel improved the MIC values against all the tested bacteria strains and decreased the hemolytic activity. Interestingly, the encapsulation protected the LL-37 peptide by the action of *P. aeruginosa* elastase [42].

### 3.4. Silica-Based Nanoparticles

Among silica NPs, Zhao and colleagues [69] investigated mesoporous one, thus observing that the inclusion of T7E21R-HD5 into MSN improved the antimicrobial activity of this defensin-derived peptide against a clinically isolated MDR *E. coli* strain [69]. In particular, the MIC value against this strain of the included peptide was four-fold lower than that of the free peptide. In addition, this complex was not cytotoxic at concentrations lower than 100 μg/mL. The authors also tested the in vivo efficacy of SCN-coating of MSN-T7E21R-HD5 through its oral administration in BALB/c mice (60 mg/Kg) infected with MDR *E. coli* [69]. Interestingly, a significant reduction of bacterial infection was observed in the ileum, cecum, and colon after treatment with MSN-T7E21R-HD5 in comparison to the untreated control group. In addition, also intestinal inflammation decreased in response to MSNs-T7E21R-HD5-SCN administration. Moreover, a reduced protein expression of tumor necrosis factor, interleukin-1β, and matrix metalloproteinase-9 was reported: these proteins are involved in the pathogenesis of enteritis caused by bacterial infection [69].

MSNs were also used as the codelivery system of multiple antibiotics to treat polymicrobial or recalcitrant infections. To this aim, Gounani and colleagues [70] evaluated the antimicrobial activity of two combined antibiotics, namely polymyxin B and vancomycin loaded at a 2:1 ratio in bare and carboxyl-modified MSNs, against *S. aureus*, *E. coli*, and *P. aeruginosa*. Specifically, at 1XMIC the antimicrobial effect was typically bacteriostatic for free antibiotics and bactericidal for antibiotics loaded in MSNs. A complete bacterial killing, instead, was observed at 4XMIC, both due to free and MSN-loaded antibiotics. However, in the case of *S. aureus* and *P. aeruginosa*, the eradication time was faster when antibiotics were loaded in MSNs. Interestingly, in all experiments, antibiotics loaded in modified MSNs were most effective in comparison to the use of bare NP [70]. In addition to hemolysis, the cytotoxic activity of MSN-loaded antibiotics was analyzed against three cell lines, HepG2, human foreskin fibroblasts (HFF-1), and human embryonic kidney cells (HEK-293). Although both free and loaded antibiotics were not hemolytic, they resulted cytotoxic as the tested concentration increased, but not in the case of antibiotics loaded in modified MSNs, which did not affect the cell viability [70]. These results indicate that modified MSNs improve the biocompatibility of polymyxin B and vancomycin, thus favoring their synergistic action.

Recently, Arquè et al. tested micro- and nanomotors, delivering LL-37 and K7-Pol peptides, against ESKAPE bacteria (*A. baumannii* AB177, *E. coli* ATCC11775, *K. pneumoniae* ATCC13883, *P. aeruginosa* PAO1, and *S. aureus* ATCC12600) [71]. Interestingly, in all cases, the functionalization of micro- and nanomotors with the two peptides increased their antimicrobial activity. Interestingly, the authors tested all the complexes in a model of mouse abscess infected with *A. baumannii*. After 2 h from infection, the mice were treated with free and micro- and nanomotors encapsulated peptides at 2XMIC concentration. The results showed that LL-37–micromotors and K7-Pol–nanomotors were the most active complexes in the presence of urea. Specifically, free peptides reduced the bacterial load only near the site of administration. In contrast, the nano- and micromotors functionalized with peptides in the presence of urea showed antibacterial activity even distant from the administration site. Moreover, no complex was found to be toxic for mice. These results suggest the importance of nano- and micromotors as new NPs for in vivo delivery of AMPs and the treatment of infective diseases [71].

### 3.5. Metal Nanoparticles

In parallel to the encapsulation with chitosan-based NPs, Sharma, et al. [38] also evaluated the activity of Pep-H loaded into AuNP against peripheral blood mononuclear cells infected by *M. tuberculosis* [38]. This nanoformulation resulted neither cytotoxic nor hemolytic. Interestingly, mycobacterial growth in monocyte-derived macrophages was reduced by about 91% when the peptide (at a concentration of 1 μg/mL) was included within AuNP and only by 45% when it was in free form, used at the same concentration [38].

Moreover, Lee et al. [74] evaluated the activity against *Vibrio vulnificus* of the synthetic peptide HPA3P^His^, loaded into AuNP conjugated to DNA aptamers (AuNP-Apt). The authors first demonstrated that the peptide was able to penetrate into HeLa cells only if encapsulated into AuNP-Apt [74]. Then, they investigated the ability of HPA3P^His^-AuNP-Apt to eradicate *V. vulnificus* from the HeLa cells previously infected. The results pointed out that AuNP-Apt improves the permeability of the peptide on mammalian cell membranes, favoring its efficacy. The antimicrobial activity of HPA3P^His^-AuNP-Apt was also confirmed in an in vivo experiment using mice septicemia models. At 42 h from the infection, all mice treated with free peptide or AuNP-Apt died. On the contrary, all mice treated with HPA3P^His^-AuNP-Apt survived up to 120 h, without significant colonization of *V. vulnificus* in the studied organs and tissues [74].

## 4. Conclusions

The infections caused by MDR pathogens are emerging as a major problem, and in many countries, National Health Services are worried about the possibility that a so-called “superbug” could become in the future a new pandemic problem like COVID-19. For this reason, the development of new antibiotics is of high importance. However, only a few have been approved in the last years, and this research sector is not considered fundamental for the pharmaceutical industry. Therefore, many countries have arranged specific guidelines and action plans related to the right use of available antibiotics and the finding of new biomedical solutions for AMR. AMPs are considered good candidates for killing pathogens due to their specific actions against biological membranes, although their therapeutic value is limited by low stability and, often, high toxicity. In this review, we have presented different nanocarriers used to encapsulate AMPs (a process sometimes called nanoencapsulation) with the specific aim of improving their potential pharmacological applications. As evidenced in the previous sections, nanoencapsulation could be performed by an active or passive targeting-based system [75,76]. The active is based on the addition of a ligand that should specifically guide the AMPs to their site of action (the external membrane of the parasite), whereas the passive is based on the encapsulation with carriers that do not modify the surface characteristics of the AMPs. Both delivery systems aim at improving stability and biocompatibility and, at the same time, at reducing toxicity and prolonging the release of AMPs. Different examples have been proposed of innovative nanoformulations that have improved AMP action against bacterial pathogens both in in vitro and in vivo studies. However, some problems still need to be addressed to quicken the progress toward the production of new antimicrobials that could rapidly go to the market. The production techniques of these nanoformulations are challenging and costly and it is not easy to find a general rule regarding the choice of the best formulation for the different identified AMPs. Some authors have recently proposed the use of nanostructures made by peptide building blocks through the self-assembling of AMPs by protein engineering, resulting in the production of a fully biocompatible and biodegradable material to avoid the problem of toxicity at higher doses [77]. Finally, the number of studies in which AMPs with nanocarriers have been employed in clinical trials is, at the moment, quite low, and, therefore, we still need to deeply understand the physiological and immunological aspects that are related to the in vivo use of these new potential drugs [78].

## Figures and Tables

**Figure 1 antibiotics-12-00184-f001:**
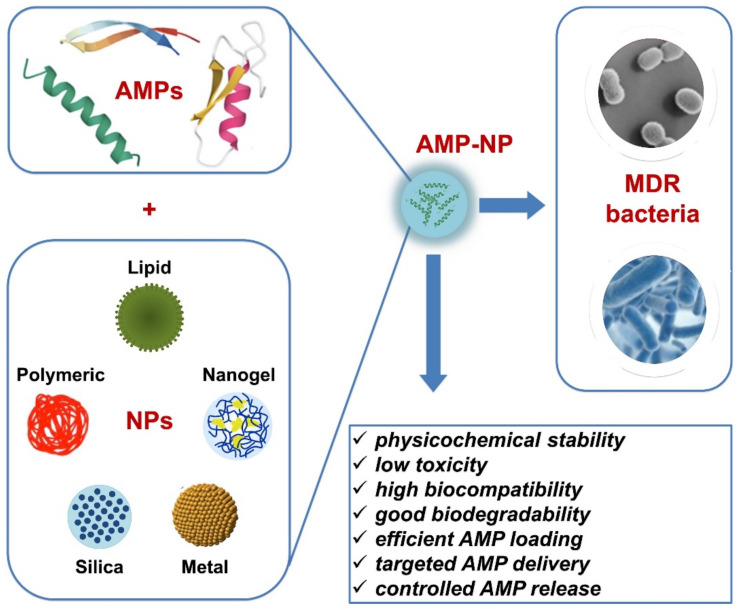
Overview of nanosystems for AMP delivery, such as lipid-based, polymeric, silica-based, metal, and nanogels. Once encapsulated, the AMPs (AMP-NP) displayed not only antimicrobial activities against MDR pathogens, but also improved properties in terms of stability, toxicity, biocompatibility, and biodegradability. The selected system will need also to assure efficient AMP loading, AMP targeted delivery, and AMP controlled release.

## Data Availability

Not applicable.

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
