# Peer review of "Antimicrobial Peptides against Bacterial Pathogens: Innovative Delivery Nanosystems for Pharmaceutical Applications"

_antibiotics, 2023, doi:10.3390/antibiotics12010184_

Round 1

Reviewer 1 Report

The manuscript of “Antimicrobial peptides against bacterial pathogens: innovative 2 delivery nanosystems for pharmaceutical applicationsis interesting to read and appreciated for good attempt. Author explained about several things like, , in vivo stability and bioavailability, by reducing 15 the eventual toxicity. I decided minor revision and also correct the following suggestion before accept the manuscript.

1.      The abstract did not fulfill all the results and it should be written corrected with obtained results.

2.      Hypothesis of introduction part is very limited. Need current explanation

3.      There are some typo and grammatical error present in the manuscript

4.      Many references are old like ten years ago, try t to add new references

5.      I don’t know why the authors are not included the figures? If manuscript having figures good for authors.

Author Response

Reviewer 1

The manuscript of “Antimicrobial peptides against bacterial pathogens: innovative delivery nanosystems for pharmaceutical applications” is interesting to read and appreciated for good attempt. Author explained about several things like in vivo stability and bioavailability, by reducing the eventual toxicity. I decided minor revision and also correct the following suggestion before accept the manuscript.

  1. The abstract did not fulfill all the results and it should be written corrected with obtained results.

We agree with this comment and the abstract was correspondingly corrected (Page 1, lines 19-25).

  1. Hypothesis of introduction part is very limited. Need current explanation.

We thank the reviewer for this comment, and we modified the introduction as requested (Page 2, lines 58-65).

  1. There are some typo and grammatical error present in the manuscript.

We apologize for typo and grammatical errors and the text was accordingly edited.

  1. Many references are old like ten years ago, try to add new references.

We revised bibliography and added new references (no. 4, 8, 14, 46, 49, 63, 68). Accordingly, we modified the reference numbering through the text and in the Table 1.

  1. I don’t know why the authors are not included the figures? If manuscript having figures good for authors.

We agree and thank the reviewer for this suggestion which has added a new dimension to our manuscript. We included the Figure 1 and its legend at page 3 (lines 82-86).

Reviewer 2 Report

First of all it was very interesting to read the manuscript and it was very well written, I would like to appreciate the effort of the authors. As a suggestion I would like to suggest kindly make few changes:

1) Line 39 to 42 : rephrase the sentence

2) Line 173 - 174: It should be removed

3) English grammatical mistakes throughout the manuscript should be checked

4) More citation if available should be added

Author Response

Reviewer 2

First of all it was very interesting to read the manuscript and it was very well written, I would like to appreciate the effort of the authors. As a suggestion I would like to suggest kindly make few changes:

  • Line 39 to 42: rephrase the sentence.

We agree with this comment and the sentence was rephrased (lines 46-46 in the revised version).

  • Line 173 - 174: It should be removed.

We apologize for this typo error and the corresponding sentence was removed.

  • English grammatical mistakes throughout the manuscript should be checked.

We apologize for grammatical mistakes and the manuscript was accordingly edited.

  • More citation if available should be added.

We thank for this suggestion and added new citations (no. 4, 8, 14, 46, 49, 63, 68). Accordingly, we modified the reference numbering through the text and in the Table 1.

Reviewer 3 Report

The manuscript reviews recent progress in properties and applications of nanomaterials used for AMP delivery. The whole review is well written, the organization of the manuscript is correct so the reader can easily follow the main discussion along the text. I think this manuscript offers a valuable and detailed view on the contemporary research on such intriguing materials and, as importantly, it may encourage and stimulate the design of new systems for useful future applications. It meets the requirements of the Antibiotics journal and therefore I recommend its publication in current form.

Author Response

We thank the reviewer for this positive comment.